# The Combination of Afatinib and Bevacizumab in Untreated EGFR-Mutated Advanced Lung Adenocarcinoma: A Multicenter Observational Study

**DOI:** 10.3390/ph13110331

**Published:** 2020-10-23

**Authors:** Ping-Chih Hsu, Chun-Yao Huang, Chin-Chou Wang, Scott Chih-Hsi Kuo, Chia-Hsun Chu, Pi-Hung Tung, Allen Chung-Cheng Huang, Chih-Liang Wang, Li-Chung Chiu, Yueh-Fu Fang, Cheng-Ta Yang

**Affiliations:** 1Department of Thoracic Medicine, Chang Gung Memorial Hospital at Linkou, Taoyuan City 33305, Taiwan; 8902049@adm.cgmh.org.tw (P.-C.H.); chihhsikuo@gmail.com (S.C.-H.K.); b9502008@cgmh.org.tw (C.-H.C.); seintseint@gmail.com (P.-H.T.); mr0818@cgmh.org.tw (A.C.-C.H.); wang@cgmh.org.tw (C.-L.W.); cremaster54@yahoo.com.tw (L.-C.C.); dr.fang.yf@gmail.com (Y.-F.F.); 2Department of Thoracic Medicine, New Taipei Municipal Tu Cheng Hospital, Chang Gung Memorial Hospital and Chang Gung University, New Taipei City 23652, Taiwan; 3Department of Pulmonary and Critical Care, Buddhist Tzu Chi General Hospital, Taipei Branch, New Taipei City 231405, Taiwan; Huang6693@yahoo.com.tw; 4Division of Pulmonary & Critical Care Medicine, Kaohsiung Chang Gung Memorial Hospital, Kaohsiung City 83301, Taiwan; ccwang5202@yahoo.com.tw; 5Department of Internal Medicine, Taoyuan Chang Gung Memorial Hospital, Taoyuan City 33378, Taiwan; 6Department of Respiratory Therapy, College of Medicine, Chang Gung University, Taoyuan City 33302, Taiwan

**Keywords:** afatinib, bevacizumab, anti-angiogenesis, epidermal growth factor receptor (EGFR), tyrosine kinase inhibitor (TKI), lung adenocarcinoma

## Abstract

The efficacy of afatinib in combination with bevacizumab in untreated advanced epidermal growth factor receptor (EGFR)-mutated lung adenocarcinoma is currently unclear. We sought to investigate the efficacy of this combination through a multicenter observational analysis. Data for 57 patients with advanced EGFR-mutated lung adenocarcinoma who received afatinib combined with bevacizumab as first-line therapy at the Chang Gung Memorial Hospitals in Linkou and Kaohsiung and Taipei Tzu Chi Hospital from May 2015 to July 2019 were analyzed. The objective response rate and disease control rate of afatinib combined with bevacizumab therapy were 87.7% and 100%, respectively. In all patients, the median progression-free survival (PFS) and overall survival (OS) were 23.9 (95% confidence interval (CI) (17.56–29.17)) and 45.9 (95% CI (39.50–53.60)) months, respectively. No statistical significance between exon 19 deletion and L858R mutations was noted in PFS or OS. The most frequent adverse events (AEs) were diarrhea (98.2%) and dermatitis (96.5%), and most AEs were grade 2 or lower and manageable. The combination of afatinib and bevacizumab is an effective therapy for untreated advanced EGFR-mutated lung adenocarcinoma with acceptable safety. Future prospective studies focusing on this combination for untreated advanced EGFR-mutated lung adenocarcinoma are warranted.

## 1. Introduction

Oncogenic driver mutations activate mutations in the epidermal growth factor receptor (EGFR) kinase domain, which occur most frequently in non-small cell lung cancer (NSCLC). The frequency of EGFR mutations in East Asians with NSCLC ranges from 45 to 55% [1,2,3]. Tyrosine kinase inhibitors (TKIs) targeting EGFR mutations have been developed for the treatment of patients with advanced lung adenocarcinoma harboring EGFR mutations, such as in exon 19 (in-frame deletions) and L858R (leucine-to-arginine substitution at codon858 point mutation in exon 21) [3,4,5]. Afatinib is a second-generation EGFR-TKI, which exhibits irreversible covalent binding to the tyrosine kinase domain of pan-ErbB receptors [5,6]. Afatinib has been approved for the treatment of EGFR-mutated advanced NSCLC because of its promising efficacy (specifically, a response rate of 55–70% and mean progression-free survival (PFS) of 11 months), which has been demonstrated in several pivotal clinical trials (LUX-Lung 3, 6, and 7 trials) [6,7,8]. Therefore, afatinib is widely used as first-line therapy for EGFR-mutated advanced NSCLC in current clinical practice.

Pathological angiogenesis has been reported to promote cancer progression in NSCLC, and anti-angiogenesis therapy has been developed and investigated in the treatment of advanced NSCLC [9,10]. Bevacizumab is a humanized monoclonal antibody that targets the ligands of vascular endothelial growth factor (VEGF) transmembrane receptors [9,10]. Bevacizumab is the first anti-angiogenesis inhibitor approved for the treatment of metastatic non-squamous lung cancer, according to the results of two previous clinical trials (E4599 and AVAIL trials) [11,12]. Previous clinical studies have shown that bevacizumab in combination with chemotherapy or EGFR-TKIs provides a survival benefit for patients with advanced lung cancer [13,14]. Combinations of first-generation EGFR-TKIs, including gefitinib and erlotinib, with bevacizumab in EGFR-mutated advanced NSCLC have been explored in clinical trials (JO25567 and NEJ026 trials), and the combination has been found to yield significantly longer PFS than EGFR-TKIs alone [15,16,17]. The efficacy of second-generation EGFR-TKI afatinib combined with bevacizumab as first-line therapy for patients with advanced EGFR-mutated NSCLC has been recently investigated in a clinical trial, but the results are not clear [18].

Here, we sought to conduct a multicenter, real-world clinical data analysis of the efficacy of afatinib in combination with bevacizumab in patients with untreated EGFR-mutated advanced lung adenocarcinoma.

## 2. Results

### 2.1. Patient Baseline Demographic Characteristics

Data for 57 patients with stage IIIB/IV lung EGFR-mutated adenocarcinoma treated between May 2015 and July 2019 were retrieved and analyzed, and the baseline characteristics and treatment information of these patients are shown in Table 1. All of the patients received afatinib combined with bevacizumab as their first-line therapy. Among them, 48 (84.2%) of the patients took afatinib at a starting dose of 40 mg/day, and nine (15.8%) had a starting dose of 30 mg/day. Among the 48 patients with the higher starting dose, 17 had dose de-escalation from 40 to 30 mg/day. Regarding the dose of bevacizumab administered, only one patient (1.8%) received 15 mg/kg in each cycle, and the other 56 patients (98.2%) received 7.5 mg/kg in each cycle. At the date of the last follow-up, the combined afatinib and bevacizumab treatment was ongoing in 27 patients (47.4%) and discontinued in 30 patients (52.6%). The combination therapy was discontinued in 28 patients because of progressive disease (PD) and in two patients because of adverse events (AEs).

### 2.2. Treatment Response, Progression-Free Survival (PFS), and Overall Survival (OS) of Afatinib Combined with Bevacizumab

Among the 57 patients included in this study, 50 (87.7%) had a partial response (PR) to afatinib combined with bevacizumab, seven (12.3%) had stable disease (SD), and none had PD at the initial evaluation. The overall objective response and disease control rates are shown in Table 2.

For all 57 patients, the median PFS of the first-line afatinib combined with bevacizumab was 23.9 months (95% confidence interval (CI), 17.56–29.17; Figure 1A), and the median OS was 45.9 months (95% CI, 39.50–53.60; Figure 1B). When the patients were divided into two groups according to their EGFR mutation, the median PFS duration of the first-line afatinib combined with bevacizumab was 23.4 and 23.9 months (95% CI, 0.71–1.69, *p* = 0.460; Figure 1C), and the median OS duration was 45.0 and 45.9 months for the L858R mutation and exon 19 deletion groups (95% CI, 0.62–1.34, *p* = 0.301; Figure 1D), respectively. There was no statistically significant difference in PFS or OS for the first-line afatinib combined with bevacizumab between the two groups.

### 2.3. Subsequent Treatment after First-Line Afatinib Combined with Bevacizumab

The subsequent treatment information for the 30 patients for whom the first-line combination of afatinib and bevacizumab was discontinued is summarized in Table 3. Of those, 24 patients underwent re-biopsy for the EGFR T790M mutation test, and 13 were positive for mutations, while 11 had negative results. Among the 13 patients with EGFR T790M mutations, 11 received subsequent third-generation EGFR-TKI osimertinib therapy, one received erlotinib combined with bevacizumab, and one received supportive care. Two of the patients with EGFR T790M mutations received bevacizumab and ramucirumab in addition to osimertinib therapy. Of the four patients who received subsequent therapy with anti-programmed death 1(PD-1)/programmed death-ligand 1 (PD-L1) immune checkpoint inhibitors (ICIs), four received the four combined regimens (atezolizumab + carboplatin + paclitaxel + bevacizumab), one received pembrolizumab combined with cisplatin plus pemetrexed, and one received nivolumab combined with cisplatin plus pemetrexed. Of the remaining two patients who received anti-angiogenesis agents in subsequent therapy, one received cisplatin plus pemetrexed with bevacizumab, and one received ramucirumab combined with cisplatin plus pemetrexed. Four patients received subsequent cisplatin plus pemetrexed doublet chemotherapy, and two patients received subsequent single pemetrexed therapy. Palliative local radiation therapy was administered in seven patients, with three receiving local radiation therapy to the brain and four to the bone.

### 2.4. Combination Afatinib and Bevacizumab Treatment-Related AEs

The combination treatment-related AEs are summarized in Table 4. Among all patients included in this study, diarrhea (98.2%) was the most frequent AE, followed by skin rashes and acne (96.5%), paronychia (77.2%), and mucositis/stomatitis (63.2%).

Twelve patients (21.1%) experienced grade 1 and 2 hypertension, which could be induced by bevacizumab. Most of the AEs were limited to grades 1 and 2 and were manageable. With regard to severe AEs (grade ≥ 3), diarrhea (21.1%) was the most frequent, followed by skin rashes and acne (17.5%), paronychia (7%), and stomatitis (6.5%). Two patients experienced grade 4 AEs consisting of skin rashes, which led to the discontinuation of combination therapy. No treatment-related deaths were recorded in this study.

## 3. Discussion

The results of our study provided evidence that the combination of afatinib and bevacizumab was an effective and safe therapy for patients with untreated EGFR-mutated advanced lung adenocarcinoma. First, this combination achieved an objective response rate of 87.8% and a disease control rate of 100%. Second, it resulted in a median PFS of 23.9 months, while the median OS was 45.9 months. Most of the AEs in this study were grade 1 or 2 AEs, and even the grade 3 AEs were controllable and reversible.

First-generation EGFR-TKIs, in combination with anti-angiogenesis agents, including bevacizumab and ramucirumab, for the treatment of advanced EGFR-mutated NSCLC, have been investigated in several studies [15,16,17,19]. A previous phase II trial (Okayama Lung Cancer Study Group Trial 1001) showed that gefitinib in combination with bevacizumab as first-line therapy for advanced EGFR-mutated NSCLC resulted in a median PFS of 14.4 months [16]. However, according to the subgroup analysis of the same trial, the combination of gefitinib and bevacizumab seemed to benefit patients with exon 19 deletion (who had a median PFS of 18.0 months), but not those with L858R mutations (who had a median PFS of 9.4 months) [16]. Two previous studies (JO25567 and NEJ026) both showed that erlotinib in combination with bevacizumab resulted in a significantly longer median PFS than erlotinib alone in untreated advanced EGFR-mutated NSCLC (16–17 months vs. 9–13 months) [15,17]. In these two studies, the combination of erlotinib and bevacizumab had effective benefits in terms of PFS in patients with either exon 19 deletion or L858R mutations [15,17]. Ramucirumab, another anti-angiogenesis inhibitor, is an immunoglobulin G1 with selective binding to VEGF receptor-2 (VEGFR-2). The binding of ramucirumab and VEGFR-2 prohibits interactions between ligands of VEGFR and VEGFR-2 [10]. Ramucirumab was approved by the United States Food and Drug Administration (FDA) in 2014 for use in treating NSCLC based on the results of the REVEL phase III clinical trial [19]. A recent study (the RELAY trial) demonstrated that ramucirumab in combination with erlotinib significantly improved PFS compared to erlotinib combined with placebo (19.4 vs. 12.4 months) in patients with untreated advanced EGFR-mutated NSCLC [20]. The combination of erlotinib and ramucirumab was equally effective in patients with exon 19 deletion and patients with L858R mutations [20]. To our knowledge, the present study is the first to show the efficacy of bevacizumab in combination with afatinib as first-line therapy for advanced EGFR-mutated lung adenocarcinoma. In this study, the combination of afatinib and bevacizumab was also shown to benefit PFS for patients with exon 19 deletion or L858R mutations (with PFS duration of 23.9 and 23.4 months, respectively). In addition, patients with brain metastasis were recruited in our study, whereas such patients were excluded in the JO25567 and RELAY trials [15,20].

The combination of afatinib and bevacizumab in chemo-naïve EGFR-mutated advanced lung cancer was investigated in a previous study (Okayama Lung Cancer Study Group Trial 1404), but that study was focused on the feasibility of this combination, not its efficacy [21]. In the same study, the response rate (RR) and disease control rate (DCR) were 81.3% and 100%, respectively, and those results were compatible with the results of our study. In addition, three of five patients (60%) with a starting afatinib dose of 40 mg/day had dose de-escalation in the Okayama Lung Cancer Study Group Trial 1404 [21]. This result suggested the use of afatinib 30 mg/day combined with bevacizumab 15 mg/kg tri-weekly in the subsequent AfaBev-CS phase II trial [18]. In our study, most of the patients (56/57, 98.2%) received 7.5 mg/kg of bevacizumab, and this might explain why a lower percentage of patients with a starting afatinib dose of 40 mg/day had dose de-escalation to 30 mg/day (n = 17, 35.4%).

In all previous trials, bevacizumab was administered at a dose of 15 mg/kg [15,16,17,18], which is different from the 7.5 mg/kg dose administered to most of the patients in our study. In Taiwan, bevacizumab is not covered by the National Health Insurance for lung cancer therapy [14,22]. The cost-effectiveness of adding bevacizumab to advanced lung adenocarcinoma treatment thus depends on the physician’s judgment and the patient’s financial resources [22]. This explains why most of the patients in our study received bevacizumab at a dose of 7.5 mg/kg, not 15 mg/kg. In addition, previous studies have shown that the 7.5 mg/kg dose is as effective as the 15 mg/kg dose when used in combination with chemotherapy for the treatment of Asian patients with non-squamous NSCLC [12,23,24]. Together, the results of this study indicated that afatinib in combination with 7.5 mg/kg of bevacizumab, with adjustable doses and treatment intervals based on the physician’s judgment of the patient’s clinical condition, could be allowed as an effective therapeutic modality for untreated advanced lung adenocarcinoma harboring sensitive EGFR mutations in real-world clinical practice.

To date, there is no ideal biomarker to predict the efficacy of bevacizumab therapy in NSCLC patients. A previous study showed that NSCLC patients with high VEGF levels had a higher probability of responding to bevacizumab combined with chemotherapy compared to chemotherapy alone [25]. The VEGF expression level did not affect the prognosis of NSCLC patients who received bevacizumab combined with chemotherapy in the same study [25]. Two previous studies demonstrated that EGFR-mutated NSCLC had increased VEGF expression levels when compared with EGFR-wild-type NSCLC [26,27]. Tanaka et al. showed that patients with advanced EGFR-mutated NSCLC benefited from bevacizumab combined with cytotoxic chemotherapy [27]. In addition, in the subgroup analysis of the IMpower 150 trial, atezolizumab plus bevacizumab plus carboplatin plus paclitaxel therapy significantly benefited patients with EGFR-mutated NSCLC in both PFS and OS [28]. Taken together, these results might explain why bevacizumab combined with EGFR-TKIs improved the response rate and PFS compared to EGFR-TKIs alone in patients with EGFR-mutated NSCLC. The Okayama Lung Cancer Study Group Trial 1404 [21] and our study showed that afatinib combined with bevacizumab yielded a more than 80% response rate, whereas single afatinib therapy had a 55–70% response rate, as shown in the LUX-Lung series [6,7,8].

Increased toxicity induced by afatinib combined with bevacizumab compared to afatinib alone should be a concern. Previous studies indicated that bevacizumab increased infrequent but serious adverse events, such as bleeding and neutropenia [10,11,12,15,16,17,23,24]. The complication of bleeding induced by bevacizumab was not recorded in this study, whereas only one bleeding adverse event was recorded in the Okayama Lung Cancer Study Group Trial 1404. Currently, clinical physicians are aware of the bevacizumab-induced bleeding reported by previous studies [10,11,12,15,16,17,23,24] and avoid administering bevacizumab in patients with a risk of bleeding, including those with cardiovascular comorbidities, great vessel invasion, cavitary tumor, symptoms of hemoptysis, and history of gastrointestinal hemorrhage. This may explain why no complication of bleeding was recorded in this study. The AfaBev-CS trial also excluded patients with bleeding risk [18]. Regarding the concern of neutropenia, an increased incidence of bevacizumab-related neutropenia mainly occurred with the combination with cytotoxic chemotherapy, but not with EGFR-TKIs [10,11,12,15,16,17,23,24]. Our study showed that the addition of bevacizumab at 7.5 mg/kg increased single afatinib-related common adverse events, including the skin toxicity, diarrhea, and paronychia, reported in the LUX-Lung series [6,7,8]. Thus, 17 patients (35.4%) in this study needed dose de-escalation (40 to 30 mg/day) because of the adverse events. A previous study reported that dose de-escalation of afatinib did not affect its efficacy in treating patients with advanced lung adenocarcinoma harboring EGFR mutations [29]. Bevacizumab-related hypertension occurred in 12 patients (21.1%) in this study, whereas this adverse effect did not occur in trials with single afatinib therapy [6,7,8]. Bevacizumab-related hypertension in this study was manageable.

Given the retrospective nature of this study, there were no suitable control patients and no direct comparison between combination and single afatinib therapy. Two previous retrospective studies showed that single afatinib had PFS of about 12 months as first-line therapy for patients with advanced EGFR-mutated NSCLC, which was similar to the results shown in the LUX-Lung series (PFS of 11–13 months) [6,7,8,30,31]. The study sites of Kuan et al. and Tu et al. were in Taiwan, indicating that the populations of these two studies were similar to that of our study [30,31]. The results of our study showed that the combination of afatinib and bevacizumab had PFS of 23.9 months, suggesting that combination might be better than single afatinib therapy in terms of PFS. Thus, more studies focusing on the comparison between afatinib with bevacizumab and single afatinib therapies are warranted. There is, however, still another limitation in this study. Only East Asian patients were recruited, so whether this combination is also effective in other racial groups besides East Asians still needs to be explored.

The EGFR T790M mutation (methionine-to-threonine substitution at amino acid position 790 in exon 20) is the most frequently acquired point mutation (30–60%), leading to resistance to afatinib in NSCLC [32,33]. In this study, the occurrence rate of T790M mutation in patients who had acquired resistance to the combination of afatinib and bevacizumab was 54.2%. The BELIEF study reported that erlotinib combined with bevacizumab therapy was effective for patients with advanced NSCLC with de novo T790M mutation [34]. A previous study showed that bevacizumab combined with afatinib induced a positive conversion of T790M mutation in previously EGFR-TKI-resistant NSCLC patients without secondary T790M mutation [35]. In the same study, the T790M mutation remained positive in those patients with previously positive T790M mutation and disease progression after afatinib plus bevacizumab therapy. Together, these results indicated that the addition of bevacizumab to afatinib therapy did not alter the T790M mutation in NSCLC patients with acquired resistance to afatinib. More than half of the patients who experienced disease progression after combined afatinib and bevacizumab were eligible for third-generation EGFR-TKI osimertinib therapy.

## 4. Materials and Methods

### 4.1. Patients

This multicenter cohort study was performed under the approval of the institutional review board (IRB) (Nos. 202000137B0 and 09-X-002), and personal informed consent was waived because the subjective data were reviewed retrospectively. Data on 1125 patients with histologically or cytologically diagnosed advanced (stage IIIB/IV and recurrent metastasis) NSCLC with EGFR mutations registered in 3 medical institutions (Linkou Chang Gung Memorial Hospital, Kaohsiung Chang Gung Memorial Hospital, and Taipei Tzu Chi Hospital) from May 2015 to July 2019 were screened. The inclusion criteria were as follows: (1) patients with activating EGFR mutations; (2) patients previously not treated with any systemic therapy, including chemotherapy, targeted therapy, or immunotherapy; (3) patients who received afatinib combined with bevacizumab as first-line therapy; (4) patients who received at least 3 cycles of bevacizumab during afatinib therapy. The exclusion criteria were as follows: (1) EGFR T790M mutation; (2) bevacizumab was not administered during afatinib therapy; (3) bevacizumab was administered for fewer than 3 cycles during afatinib therapy. The inclusion and exclusion criteria for retrieving patients are shown in Figure 2.

The complete medical history, imaging studies, treatments, and treatment-related adverse effects for each patient were retrospectively retrieved from medical records. Images, including chest radiography, computed tomography (CT), brain magnetic resonance imaging (MRI), and positron emission tomography (PET) scans, were performed at baseline to determine the stage and condition of metastasis. Whole-body CT scans were performed regularly during the combined afatinib and bevacizumab therapy to evaluate the efficacy of treatment. Additional images, such as PET scans, sonograms, and MRIs, were also obtained and judged by clinical physicians as needed.

The last follow-up time point in this study was July 2020.

### 4.2. EGFR Mutation Test

Direct sequencing, as previously described [36], or amplified refractory mutation system—Scorpion [37], depending on the facilities of different institutions, was used to detect EGFR mutations.

### 4.3. Evaluation of Efficacy of Combined Afatinib and Bevacizumab Therapy

The patients were treated with afatinib at a starting dose of either 40 or 30 mg once daily until disease progression or intolerable toxicity occurred. Bevacizumab was administered intravenously with an initial dose of 15 or 7.5 mg/kg during afatinib therapy. The dose and schedule of afatinib and bevacizumab were adjusted by the physicians depending on the patient’s clinical condition and toxicity from treatment. Response Evaluation Criteria in Solid Tumors version 1.1 was used for the assessment of treatment responses, defined as PD, SD, and PR. PFS was defined as the duration from the first date of afatinib administrated until the date of the first imaging evidence of disease progression, death, or last follow-up. OS was defined as the duration from the date of diagnosis until the date of recorded mortality. If patients were still alive on 31 July 2020 (the last follow-up time point), the survival was censored at the last visit date recorded.

### 4.4. Treatment-Related AEs

Data regarding treatment-related toxicity and AEs were collected from the electronic medical records and graded by the National Cancer Institute Common Terminology Criteria for Adverse Events, version 4.0.

### 4.5. Statistical Analysis

The baseline demographic characteristics and treatment information data were presented as numbers for qualitative variables or as medians and ranges for quantitative variables as appropriate. The Kaplan–Meier method was used to calculate survival rates. The data were presented as median time estimated and 95% confidence interval (CI) for the median value. A comparison of survival curves was conducted according to a log-rank test, and all *p*-values were 2-sided, with *p* < 0.05 defined as statistically significant. All statistical analyses were performed by using GraphPad Prism, version 5.0 (GraphPad Software, San Diego, CA, USA).

## 5. Conclusions

The combination of afatinib with bevacizumab is an effective therapy for patients with untreated stage IIIB/IV EGFR-mutated lung adenocarcinoma, and the safety of this combination is acceptable. More prospective studies focusing on the combination of afatinib and bevacizumab for untreated advanced EGFR-mutated lung adenocarcinoma are warranted.

## Figures and Tables

**Figure 1 pharmaceuticals-13-00331-f001:**
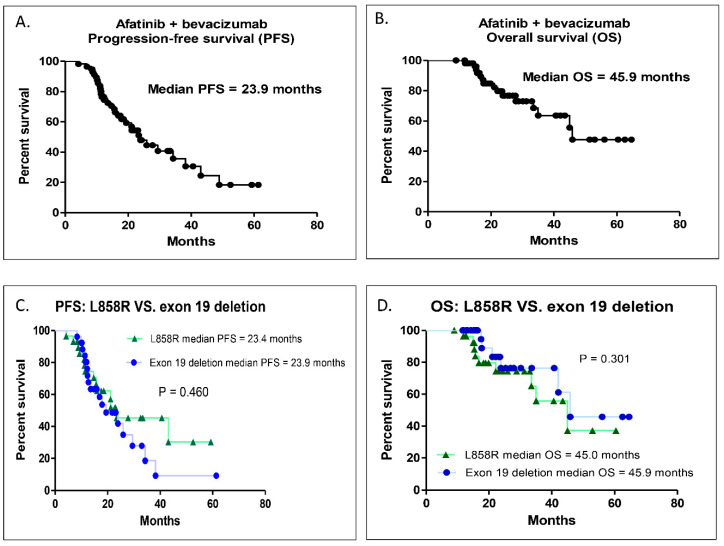
Kaplan–Meier survival curve of PFS and OS. (**A**) Overall PFS in this study. (**B**) Overall OS in this study. (**C**) Comparison of PFS between exon 19 deletion and L858R mutation groups. (**D**) Comparison of OS between exon 19 deletion and L858R mutation groups. PFS, progression-free survival; OS, overall survival.

**Figure 2 pharmaceuticals-13-00331-f002:**
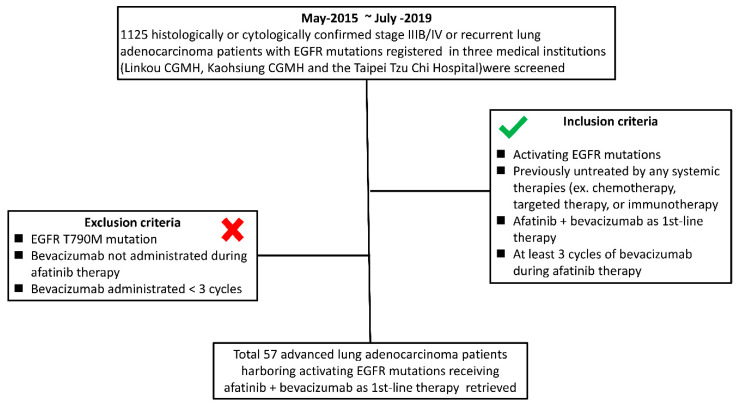
Inclusion and exclusion criteria flowchart of this study.

**Table 1 pharmaceuticals-13-00331-t001:** Baseline characteristics and treatment information of study patients. EGFR, epidermal growth factor receptor.

Total	*n* = 57
Sex	
Male/female	23/34
Age (range/median)	32–80/60
ECOG PS	
0–1	57
≥2	0
Smoking Status	
Non-smoker	43
Smoker (Current and Former)	14
Histology	
Adenocarcinoma	57
Stage	
IIIB/IV	4/53
EGFR mutation	
L858R	29
Exon 19 deletion	26
Uncommon mutation *	2
Brain metastasis at diagnosis	10
Treatment information	
Afatinib starting dose	
40 mg/day	48
30 mg/day	9
Dose de-escalation (40 mg -> 30 mg)	17
Bevacizumab dose (each cycle)	
15 mg/kg	1
7.5 mg/kg	56
Treatment ongoing	27
Treatment discontinued	30
Reason for discontinued	
Disease progression	28
Intolerant toxicity	2

ECOG PS: Eastern Cooperative Oncology Group (ECOG) Performance Status; * G719X and S768I.

**Table 2 pharmaceuticals-13-00331-t002:** Treatment response and efficacy of afatinib in combination with bevacizumab.

Total	*n* = 57
Complete response (CR)	0
Partial response (PR)	50
Stable disease (SD)	7
Progressive disease (PD)	0
Response rate (RR)%	87.7
Disease control rate (DCR) %	100
Median PFS (months)	23.9 (95% CI (17.56–29.17))
Median OS (months)	45.9 (95% CI (39.50–53.60))

PFS, progression-free survival; OS, overall survival; CI, confidence interval.

**Table 3 pharmaceuticals-13-00331-t003:** Subsequent treatment information after afatinib + bevacizumab.

	*n*
Re-biopsy for EGFR T790M mutation test	24
EGFR T790M mutation status	
Positive	13
Negative	11
Unknown (no re-biopsy or ctDNA for EGFR T790M mutation tests)	6
EGFR T790M mutation rate (%)	54.2%
Subsequent therapy	
Osimertinib	11
Erlotinib	1
Platinum-base doublet chemotherapy	10
Single agent chemotherapy (pemetrexed)	2
Anti-PD-1/PD-L1 ICIs	4
Bevacizumab	4
Ramucirumab	2
Supportive care	6
Local radiation therapy	
Brain	3
Bone	4

PD-1, programmed death 1; PD-L1, programmed death-ligand 1; ICIs, immune checkpoint inhibitors.

**Table 4 pharmaceuticals-13-00331-t004:** Treatment-related adverse events (AEs) of combined afatinib and bevacizumab therapy.

Adverse Event (AE)	All *n* = 57 (*n* (%))	Grade 1–2 (*n* (%))	Grade 3 (*n* (%))	Grade 4 (*n* (%))
Skin rash/acne	55 (96.5)	45 (78.9)	8 (14)	2 (3.5)
Diarrhea	56 (98.2)	44 (77.1)	12 (21.1)	0
Stomatitis	36 (63.2)	33 (57.9)	3 (5.3)	0
Paronychia	44 (77.2)	40 (70.2)	4 (7)	0
Nausea or vomiting	10 (17.5)	10 (17.5)	0	0
Increased liver transaminases	2 (3.5)	2 (3.5)	0	0
Hypertension	12 (21.1)	12 (21.1)	0	0

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
