# Peer review of "The Combination of Afatinib and Bevacizumab in Untreated EGFR-Mutated Advanced Lung Adenocarcinoma: A Multicenter Observational Study"

_pharmaceuticals, 2020, doi:10.3390/ph13110331_

Round 1

Reviewer 1 Report

This study retrospected the efficacy and safety of a combination of Afatinib with Bevacizumab in untreated advanced epidermal growth factor receptor (EGFR)-mutated lung adenocarcinoma patients. The study demonstrated that the combination was feasible and associated with clinical benefit and tolerable, reversible side effects. In general, this is an interesting and meaningful study. However, this study only reviewed those patients receiving the combination therapy; and the lack of analysis of control patients treated with each single regimen limited the clinical significance of the study. Therefore it seems it is hard to draw a conclusion that the combination is a better choice than a single regimen for those patients.

Major concerns

  1. How was the combination therapy compared with single regimens?
  2. How was the efficacy of Bevacinzumab against angiogenesis evaluated in clinical and also in the study? Please further address this.
  3. Did Bevacinzumab delay the occurrence of T790M mutation? The underlying rationales and mechanisms for the combination therapy should be introduced and discussed.

Author Response

Point 1: English language and style are fine/minor spell check required. ʉ۬

Response 1: The manuscript has been revised by English language editing service of MDPI.

Point 2: How was the combination therapy compared with single regimens?

Response 2: In response to the concern of how was the combination therapy compared with single regimens, this study mainly focus on the efficacy of the combination of afatinib combined with bevacizumab, not in comparison between combination and single agent.

Given the retrospective nature of this study, there were no suitable control patients included and no direct comparison between the combination and single afatinib therapy. Two previous retrospective studies showed that single afatinib had about 12 months of PFS as first-line therapy for advanced EGFR-mutated NSCLC patients, and the results were similar to that shown in LUX-Lung series (11-13 months of PFS) [1-5]. The study sites of Kuan et al. and Tu et al. located in Taiwan which indicated that the population of these 2 studies is similar to those of our study [4,5]. The result of our study showed that the combination of afatinib and bevacizumab had 23.9 months of PFS, and suggest that combination may be better than single afatinib therapy in PFS. Thus, more future studies focus on the comparison between afatinib combined with bevacizumab and single afatinib therapies are warranted.

We added this discussion paragraph was in page 8 line 233-242, and references 31 and 32 in revised manuscript.

Point 3: How was the efficacy of Bevacinzumab against angiogenesis evaluated in clinical and also in the study? Please further address this.

 Response 3: We discussed the efficacy of bevacinzumab against angiogenesis in EGFR-mutated NSCLC patients as suggested in revised manuscript.

To date, there is no ideal biomarker to predict the efficacy of bevacizumab therapy in NSCLC patients. A previous study showed that NSCLC patients with high VEGF level had increased probability of responding to bevacizumab combined with chemotherapy when compare to chemother alone [6]. The VEGF expression level did not affect the prognosis of NSCLC patient who received bevacizumab combined with chemotherapy in the same study [6]. Two previous studies demonstrated that EGFR-mutated NSCLC had increased VEGF expression level when compared with EGFR-wild type NSCLC [7,8]. Tanaka et al. showed that advanced EGFR-mutated NSCLC patients benefit by bevacizumab in combination with cytotoxic chemotherapy [8]. In addition, in the subgroup analysis of IMpower 150 trial, atezolizumab plus bevacizumab plus carboplatin plus paclitaxel therapy significantly benefit EGFR-mutated NSCLC patients in both PFS and OS [9. Taken together, these results may explain that why bevacizumab combined with EGFR-TKIs improved response rate and PFS compared to EGFR-TKIs alone in EGFR-mutated NSCLC patients. The Okayama Lung Cancer Study Group Trial 1404 and our study showed afatinib in combination with bevacizumab yield more than 80% of response rate whether single afatinib therapy had 55-70% of response rates which were shown in LUX-Lung series.

This discussion paragraph was added in page 7 line 198-217 in revised manuscript. We also add references 25-28.

Point 4: Did Bevacinzumab delay the occurrence of T790M mutation? The underlying rationales and mechanisms for the combination therapy should be introduced and discussed.

 Response 4: We discussed the point about of “did Bevacinzumab delay the occurrence of T790M mutation” as suggested in revised manuscript.

The EGFR T790M mutation (the methionine to threonine substitution at amino acid position 790 in exon 20) is the most frequent (30–60%) acquired point mutation which lead to the drug resistance to afatinib in NSCLC. In this study, the occurrence rate of T790M mutation in patients who had acquired resistance to the combination of afatinib and bevacizumab was 54.2%. The BELIEF study reported that erlotinib combined with bevacizumab therapy was effective for advanced NSCLC patients with de novo T790M mutation [10]. A previous study showed that bevacizumab combined with afatinib induced positive conversion of T790M mutation in previously EGFR-TKI resistant NSCLC patients without secondary T790M mutation [11]. In the same study, the T790M mutation remained positive in those patients with previously positive T790M mutation and disease progression after afatinib plus bevacizumab therapy. Together, these results indicate that the addition of bevacizumab to afatinib therapy dose not alter the T790M mutation in NSCLC patients with acquired resistance to afatinib. More than half of patients who experience disease progression after the combination afatinib and bevacizumab are eligible for 3rd generation EGFR-TKI Osimertinib therapy.

This discussion paragraph was added in page 8 line 245-258 in revised manuscript. We also add references 34,35.

References:

  1. Sequist, L.V.; Yang, J.C.-H.; Yamamoto, N.; Obyrne, K.; Hirsh, V.; Mok, T.; Geater, S.L.; Orlov, S.; Tsai, C.-M.; Boyer, M.; et al. Phase III Study of Afatinib or Cisplatin Plus Pemetrexed in Patients With Metastatic Lung Adenocarcinoma With EGFR Mutations. J. Clin. Oncol. 2013, 31, 3327–3334.
  2. Wu, Y.L.; Zhou, C.; Hu, C.P.; Feng, J.; Lu, S.; Huang, Y.; Li,W.; Hou, M.; Shi, J.H.; Lee, K.Y. et al. Afatinib versus cisplatin plus gemcitabine for first-line treatment of Asian patients with advanced non-small-cell lung cancer harbouring EGFR mutations (LUX-Lung 6): An open-label, randomised phase 3 trial. Lancet Oncol. 2014, 15, 213–222.
  3. Park, K.; Tan, E.H.; O'Byrne K, Zhang L, Boyer M, Mok T, Hirsh V, Yang JC, Lee KH, Lu S, Shi Y, Kim SW, Laskin J, Kim DW, Arvis CD, Kölbeck K, Laurie SA, Tsai CM, Shahidi M, Kim M, Massey D, Zazulina V, Paz-Ares L. Afatinib versus gefitinib as first-line treatment of patients with EGFR mutation-positive non-small-cell lung cancer (LUX-Lung 7): a phase 2B, open-label, randomised controlled trial. Lancet Oncol. 2016 May;17(5):577-89.
  4. Kuan, F.C.; Li, S.H.; Wang, C.L.; Lin, M.H.; Tsai, Y.H.; Yang, C.T. Analysis of progression-free survival of first-line tyrosine kinase inhibitors in patients with non-small cell lung cancer harboring leu858Arg or exon 19 deletions. Oncotarget. 2017 Jan 3;8(1):1343-1353.
  5. Tu, C.Y.; Chen, C.M.; Liao, W.C.; Wu, B.R.; Chen, C.Y.; Chen, W.C.; Hsia, T.C.; Cheng, W.C.; Chen, C.H. Comparison of the effects of the three major tyrosine kinase inhibitors as first-line therapy for non-small-cell lung cancer harboring epidermal growth factor receptor mutations. Oncotarget. 2018 Feb 4;9(36):24237-24247.
  6. Dowlati, A.; Gray, R.; Sandler, A.B.; Schiller, J.H.; Johnson, D.H. Cell adhesion molecules, vascular endothelial growth factor, and basic fibroblast growth factor in patients with non-small cell lung cancer treated with chemotherapy with or without bevacizumab-an Eastern Cooperative Oncology Group Study. Clin. Cancer Res. 2008, 14, 1407–1412.
  7. Reinmuth, N.; Jauch, A.; Xu, E.C.; Muley, T.; Granzow, M.; Hoffmann, H.; Dienemann, H.; Herpel, E.; Schnabel, P.A.; Herth, F.J.; et al. Correlation of EGFR mutations with chromosomal alterations and expression of EGFR, ErbB3 and VEGF in tumor samples of lung adenocarcinoma patients. Lung Cancer. 2008 Nov;62(2):193-201.
  8. Tanaka, I.; Morise, M.; Miyazawa, A.; Kodama, Y.; Tamiya, Y.; Gen, S.; Matsui, A.; Hase, T.; Hashimoto, N.; Sato, M.; et al. Potential Benefits of Bevacizumab Combined With Platinum-Based Chemotherapy in Advanced Non-Small-Cell Lung Cancer Patients With EGFR Mutation. Clin Lung Cancer. 2020 May;21(3):273-280.e4.
  9. Reck, M.; Mok, T.S.K.; Nishio, M.; Jotte, R.M.; Cappuzzo, F.; Orlandi, F.; Stroyakovskiy, D.; Nogami, N.; Rodríguez-Abreu, D.; Moro-Sibilot, D.; et al. Atezolizumab plus bevacizumab and chemotherapy in non-small-cell lung cancer (IMpower150): key subgroup analyses of patients with EGFR mutations or baseline liver metastases in a randomised, open-label phase 3 trial. Lancet Respir Med. 2019 May;7(5):387-401.
  10. Rosell, R.; Dafni, U.; Felip, E.; Curioni-Fontecedro, A.; Gautschi, O.; Peters, S.; Massutí, B.; Palmero, R.; Aix, S.P.; Carcereny, E.; et al. Erlotinib and bevacizumab in patients with advanced non-small-cell lung cancer and activating EGFR mutations (BELIEF): an international, multicentre, single-arm, phase 2 trial. Lancet Respir Med. 2017 May;5(5):435-444. 
  11. Hata, A.; Katakami, N.; Kaji, R.; Yokoyama, T.; Kaneda, T.; Tamiya, M.; Inoue, T.; Kimura, H.; Yano, Y.; Tamura, D.; et al. Does afatinib plus bevacizumab combination therapy induce positive conversion of T790M in previously-negative patients? Oncotarget. 2018 Oct 5;9(78):34765-34771.

Reviewer 2 Report

This study retrospectively assessed the clinical efficacies of afatinib plus bevacizumab (7.5 mg/kg) in untreated advanced EGFR-mutated lung adenocarcinoma. The data was collected from multicenter, and the combination therapy showed the median PFS of 23.9 months and the median OS of 45.9 moths with acceptable safety. The paper firstly indicates the clinical benefits of the combination therapy in the real-world setting, and provide us clinically meaningful evidence. The results were well written, and the reviewer has few comments as follows:

1.The combination therapies with anti-angiogenetic inhibitors, bevacizumab, have increased the risk of infrequent serious adverse events, such as bleeding events and neutropenia complications. The present study showed the acceptable toxicities of the addition of bevacizumab at the dose of 7.5 mg/kg. It will be useful to discuss the comparison of the adverse events between afatinib with bevacizumab and afatinib alone, including the dose setting.

2.  There are no valid predictive biomarkers of response to treatment with bevacizumab, while several papers showed VEGFA expression was increased in EGFR-mutant NSCLC compared with EGFR-wild type (Lung Cancer. 2008 Nov;62(2):193-201. Clin Lung Cancer. 2020 May;21(3):273-280.e4.). Moreover, in the subgroup analysis of IMpower 150 trial, atezolizumab plus bevacizumab plus carboplatin plus paclitaxel (ABCP) therapy significantly improved PFS for EGFR-mutated NSCLC patients compared to bevacizumab plus carboplatin plus paclitaxel (BCP) therapy, suggesting that the addition of bevacizumab to any other drugs could be beneficial for NSCLC with EGFR mutations. The reviewer recommends to indicate the papers and discuss about anti-angiogenetic strategies in patients with EGFR mutations.

Author Response

Point 1: Moderate English changes required

Response 1: The manuscript has been revised by English language editing service of MDPI.

Point 2: The combination therapies with anti-angiogenetic inhibitors, bevacizumab, have increased the risk of infrequent serious adverse events, such as bleeding events and neutropenia complications. The present study showed the acceptable toxicities of the addition of bevacizumab at the dose of 7.5 mg/kg. It will be useful to discuss the comparison of the adverse events between afatinib with bevacizumab and afatinib alone, including the dose setting.

 Response 2: We added a paragraph to discuss the adverse events between afatinib with bevacizumab and afatinib alone as suggested.

      Increased toxicities induced by the afatinib combined with bevacizumab compared to afatinib alone should be concerned. Previous studies indicated that bevacizumab increased infrequent but serious adverse events such as bleeding and neutropenia. The complication of bleeding induced by bevacizumab was not recorded in this study, whether only one bleeding adverse event recorded in the Okayama Lung Cancer Study Group Trial 1404. Currently, clinical physicians aware of the bevacizumab-induced bleeding reported by previous studies, and avoid administrating of bevacizumab in patients with risk of bleeding including cardiovascular comorbidities, great vessel invasion, cavitary tumor, symptom of hemoptysis and history of gastrointestinal hemorrhage. This may explain that why there is no complication of bleeding recorded in this study. Therefore, the AfaBev-CS trial also excluded patients with risks of bleeding . Regarding the concern of neutropenia, increased incidence of bevacizumab-related neutropenia mainly occurred in the combination with cytotoxic chemotherapy but not in with EGFR-TKIs. Our study showed that the addition of bevacizumab in dose of 7.5mg/kg increased single afatinib-related common adverse events including skin toxicity, diarrhea and paronychia reported in LUX-Lung series [6-8]. Thus, 17 (35.4%) patients in this study needed dose de-escalation (40mg/day to 30mg/day) because of the adverse events. A previous study reported that dose de-escalation of afatinib do not affect its efficacy in treating advanced lung adenocarcinoma patients harboring EGFR mutations. Bevacizumab-related hypertension occurred in 12 (21.1%) patients of this study whether this adverse effect did not appear in trials with single afatinib therapy. The bevacizumab-related hypertension in this study were all manageable.

This discussion paragraph was added in page 7 line 213-232 in revised manuscript

Point 3: There are no valid predictive biomarkers of response to treatment with bevacizumab, while several papers showed VEGFA expression was increased in EGFR-mutant NSCLC compared with EGFR-wild type (Lung Cancer. 2008 Nov;62(2):193-201. Clin Lung Cancer. 2020 May;21(3):273-280.e4.). Moreover, in the subgroup analysis of IMpower 150 trial, atezolizumab plus bevacizumab plus carboplatin plus paclitaxel (ABCP) therapy significantly improved PFS for EGFR-mutated NSCLC patients compared to bevacizumab plus carboplatin plus paclitaxel (BCP) therapy, suggesting that the addition of bevacizumab to any other drugs could be beneficial for NSCLC with EGFR mutations. The reviewer recommends to indicate the papers and discuss about anti-angiogenetic strategies in patients with EGFR mutations.

Response 3: We discussed the anti-angiogenetic strategies in patients with EGFR mutations and cited the references as suggested in revised manuscript.

To date, there is no ideal biomarker to predict the efficacy of bevacizumab therapy in NSCLC patients. A previous study showed that NSCLC patients with high VEGF level had increased probability of responding to bevacizumab combined with chemotherapy when compare to chemother alone [1]. The VEGF expression level did not affect the prognosis of NSCLC patient who received bevacizumab combined with chemotherapy in the same study [1]. Two previous studies demonstrated that EGFR-mutated NSCLC had increased VEGF expression level when compared with EGFR-wild type NSCLC [2,3]. Tanaka et al. showed that advanced EGFR-mutated NSCLC patients benefit by bevacizumab in combination with cytotoxic chemotherapy [2]. In addition, in the subgroup analysis of IMpower 150 trial, atezolizumab plus bevacizumab plus carboplatin plus paclitaxel therapy significantly benefit EGFR-mutated NSCLC patients in both PFS and OS [4]. Taken together, these results may explain that why bevacizumab combined with EGFR-TKIs improved response rate and PFS compared to EGFR-TKIs alone in EGFR-mutated NSCLC patients. The Okayama Lung Cancer Study Group Trial 1404 and our study showed afatinib in combination with bevacizumab yield more than 80% of response rate whether single afatinib therapy had 55-70% of response rates which were shown in LUX-Lung series.

This discussion paragraph was added in page 7 line 198-212 in revised manuscript

We also added references 25-28

References:

  1. Dowlati, A.; Gray, R.; Sandler, A.B.; Schiller, J.H.; Johnson, D.H. Cell adhesion molecules, vascular endothelial growth factor, and basic fibroblast growth factor in patients with non-small cell lung cancer treated with chemotherapy with or without bevacizumab-an Eastern Cooperative Oncology Group Study. Clin. Cancer Res. 2008, 14, 1407–1412.
  2. Reinmuth, N.; Jauch, A.; Xu, E.C.; Muley, T.; Granzow, M.; Hoffmann, H.; Dienemann, H.; Herpel, E.; Schnabel, P.A.; Herth, F.J.; et al. Correlation of EGFR mutations with chromosomal alterations and expression of EGFR, ErbB3 and VEGF in tumor samples of lung adenocarcinoma patients. Lung Cancer. 2008 Nov;62(2):193-201.
  3. Tanaka, I.; Morise, M.; Miyazawa, A.; Kodama, Y.; Tamiya, Y.; Gen, S.; Matsui, A.; Hase, T.; Hashimoto, N.; Sato, M.; et al. Potential Benefits of Bevacizumab Combined With Platinum-Based Chemotherapy in Advanced Non-Small-Cell Lung Cancer Patients With EGFR Mutation. Clin Lung Cancer. 2020 May;21(3):273-280.e4.
  4. Reck, M.; Mok, T.S.K.; Nishio, M.; Jotte, R.M.; Cappuzzo, F.; Orlandi, F.; Stroyakovskiy, D.; Nogami, N.; Rodríguez-Abreu, D.; Moro-Sibilot, D.; et al. Atezolizumab plus bevacizumab and chemotherapy in non-small-cell lung cancer (IMpower150): key subgroup analyses of patients with EGFR mutations or baseline liver metastases in a randomised, open-label phase 3 trial. Lancet Respir Med. 2019 May;7(5):387-401.
  5. Ninomiya, T.; Nogami, N.; Kozuki, T.; Harada, D.; Kubo, T.; Ohashi, K.; Kuyama, S.; Kudo, K.; Bessho, A.; Fukamatsu, N.; et al. A phase I trial of afatinib and bevacizumab in chemo-naïve patients with advanced non-small-cell lung cancer harboring EGFR mutations: Okayama Lung Cancer Study Group Trial 1404. Lung Cancer. 2018 Jan;115:103-108.